# Phyloecology of nitrate ammonifiers and their importance relative to denitrifiers in global terrestrial biomes

Aurélien Saghaï [1], Grace Pold[1], Christopher M. Jones[1] & Sara Hallin [1] ✉

Nitrate ammonification is important for soil nitrogen retention. However, the ecology of ammonifiers and their prevalence compared with denitrifiers, being competitors for nitrate, are overlooked. Here, we screen 1 million genomes for *nrfA and onr*, encoding ammonifier nitrite reductases. About 40% of ammonifier assemblies carry at least one denitrification gene and show higher potential for nitrous oxide production than consumption. We then use a phylogeny-based approach to recruit gene fragments of *nrfA, onr* and denitrification nitrite reductase genes (*nirK*, *nirS*) in 1861 global terrestrial metagenomes. *nrfA* outnumbers the nearly negligible *onr* counts in all biomes, but denitrification genes dominate, except in tundra. Random forest modelling teases apart the influence of the soil C/N on *nrfA*-ammonifier vs denitrifier abundance, showing an effect of nitrate rather than carbon content. This study demonstrates the multiple roles nitrate ammonifiers play in nitrogen cycling and identifies factors ultimately controlling the fate of soil nitrate.

Human activity, in particular agricultural fertilizer application and fossil-fuel combustion, has increased the amount of nitrogen (N) circulating in the biosphere and created an imbalance in the N cycle that threatens ecosystem integrity at the global scale[1]. Nitrate is a highly mobile form of reactive N in soil and the primary source of global N pollution[2]. If not assimilated into biomass or leached to watersheds, nitrate can be used as an electron acceptor by soil microorganisms under oxygen-limited conditions, mainly through denitrification or nitrate ammonification, also known as dissimilatory nitrate reduction to ammonium. Denitrification leads to N loss through the production of gaseous N-compounds, including the potent greenhouse gas nitrous oxide ($N_2O$). Terrestrial ecosystems contribute ca. 60% to global $N_2O$ emissions[3], with denitrification being the main source, and $N_2O$ concentration in the atmosphere is increasing at an accelerating rate[4]. By contrast, only small amounts of $N_2O$ have been detected from isolates performing nitrate ammonification[5–7] and the process results in the retention of N via the binding of ammonium to negatively charged surfaces in the soil. Determining the environmental factors controlling the end-products when nitrate is used as electron acceptor is crucial for our ability to predict and influence N budgets in terrestrial ecosystems at the global scale[8]. A key factor is a better understanding of the ecology of nitrate-

ammonifying microorganisms, as they are an overlooked functional group in the N cycle[9], particularly in terrestrial ecosystems.

Nitrate ammonification is primarily driven by microorganisms using the pentaheme cytochrome c nitrite reductase NrfA, encoded by the *nrfA* gene, to catalyze the reduction of nitrite to ammonium[10]. Microorganisms can also catalyze this reaction using the octaheme nitrite reductase (ONR), a close homolog of NrfA encoded by the *onr* gene[11]. However, its prevalence and importance for ammonification in the environment has been less studied compared to *nrfA*. Similar to denitrification, nitrate is reduced to nitrite by reductases encoded by *narG* or *napA*, and the branching point between the two pathways is the reduction of nitrite. While nitrate ammonification involves the reduction of nitrite to ammonium in a single enzymatic step, denitrification is a modular pathway in which nitrite is successively reduced into nitric oxide, $N_2O$ and finally dinitrogen gas via reductases encoded by *nirK* or *nirS*, *nor* and *nosZ*, respectively[12]. Both nitrate ammonification driven by NrfA and the competing process denitrification are performed by phylogenetically diverse bacteria and archaea[13,14], whereas ammonification involving ONR is constrained to Proteobacteria[11]. Ammonification has been suggested to dominate or increase in relation to denitrification under electron acceptor

[1]Swedish University of Agricultural Sciences, Department of Forest Mycology and Plant Pathology, Uppsala, Sweden. ✉e-mail: sara.hallin@slu.se

limitation, i.e. at high ratios of soil organic carbon (SOC) and nitrate, whereas conditions with electron donor limitation favor denitrification[15]. This is supported by more recent work with enrichment and pure cultures[5,16–18], site-specific field studies[19–21] and modeling approaches[22,23]. Yet, we lack a synthesis of the relative importance of the SOC to nitrate ratio and other factors for the competition to use nitrate as an electron acceptor across Earth's terrestrial biomes[24].

In this study, we determine the extant diversity, abundance, and global distribution of ammonifiers and the environmental drivers of the potential competition with denitrifiers in terrestrial ecosystems. This includes an extensive phylogenetic analysis of full-length *nrfA* and *onr* sequences obtained from screening more than 1,000,000 assemblies of isolate and metagenome-assembled genomes (MAGs). Both ammonification and denitrification have been shown to co-exist in a few but phylogenetically diverse isolates[18,25,26]. Therefore, the presence of denitrification genes in the assemblies was determined to gain insights into the overall patterns of N$_2$O production and reduction capacity among ammonifiers. In addition to the *nrfA/onr* phylogeny, we also use updated phylogenies of the genes *nirK* and *nirS*[12], coding for the equivalent function in denitrifiers, to provide a phylogenetic framework for analyzing ammonifying and denitrifying microorganisms. The phylogenies were used as references for recruiting bacterial and archaeal *nrfA, onr, nirK*, and *nirS* fragments from 1861 soil and rhizosphere metagenomes derived from broad environmental gradients across 725 locations to assess the global distribution of ammonifiers and their abundance relative to denitrifying microorganisms. Because *onr* counts were very low across all metagenomes, we identified environmental drivers underpinning the abundance of functional microbial communities performing NrfA-driven ammonification *vs.* denitrification as a proxy for the competition between these groups using random forest modelling and discuss the implications of the findings for N loss and retention in global soils.

## Results

### Phylogeny of NrfA and ONR

The search for the presence of *nrfA* and *onr* in isolate genomes and MAGs resulted in 1155 and 106 non-redundant taxonomically and structurally diverse sequences, respectively. Overall, 1113 and 93 assemblies carried *nrfA* and *onr* alone, respectively, whereas 12 carried both. The *nrfA* diversity spanned 1 archaeal and 44 bacterial phyla, whereas *onr* sequences were detected in 1 archaeal and 14 bacterial phyla, respectively (Supplementary Table 1). Only 25 *nrfA*-assemblies and 1 *onr*-assembly carried more than one copy of *nrfA* and *onr*, respectively, with high sequence similarity between copies (Supplementary Fig. 1).

Phylogenetic reconstruction confirmed that NrfA and ONR sequences form distinct and monophyletic clades[27]. The NrfA region of the tree was overall congruent with that of the organisms at the class level, except for some taxa including Anaerolineae, Campylobacteria, Gammaproteobacteria and Myxococcia (Fig. 1). While all sequences contained the five heme-binding sites and a histidine residue between the third and fourth site that are characteristic of NrfA[28], a number of other structural features were associated with different clades within the NrfA region in the phylogeny. Sequences with a Cys-X-X-Cys-His (CXXCH) motif in the first site, instead of the more common Cys-X-X-Cys-Lys, were exclusively bacterial and formed a monophyletic, well supported and taxonomically diverse clade (Fig. 1)[14,27]. Most known NrfA proteins are characterized by the presence of a calcium ion near the active site, where it is suspected to play a structural role[29], whereas those that are calcium-independent contain a X-X-Arg-His motif between the third and fourth sites. The latter were present in several regions of the tree (*n* = 131 sequences), supporting independent evolutionary events[30]. By contrast, all ONR proteins were calcium-dependent and the sequences displayed eight heme-binding sites (3x-CXXCH-1x-CXXCK-4x-CXXCH), with a histidine residue between the sixth and seventh site[11].

### Potential for denitrification in genomes of *nrfA*- and *onr*-ammonifiers

The 1218 genomes harboring *nrfA* and/or *onr* were further examined for the presence of denitrification genes. About 42% of the assemblies harboring *nrfA* but not *onr* contained at least one denitrification gene (*nir, nor* or *nosZ*), whereas 13% carried more than one denitrification gene, with complete denitrifiers accounting for just 2.5% of the *nrfA*-ammonifiers (Fig. 2a; Supplementary Table 2). These proportions were comparatively higher in the CXXCH clade, except for complete denitrifiers (60%, 18% and <1%, respectively). Among *nrfA*-assemblies with at least one denitrification gene, about 50% were potential N$_2$O producers, carrying *nor* but not *nosZ*, whereas only 38% were potential N$_2$O consumers, carrying *nosZ* alone or in addition to *nir/nor* (51 and 31% in the CXXCH clade, respectively). Regarding the *onr*-encoding assemblies, either alone or in combination with *nrfA*, 40% carried at least one denitrification gene, with the potential for N$_2$O consumption limited to a few gammaproteobacterial genomes (Fig. 2c, e). Overall, this suggests a higher genetic potential for N$_2$O production than consumption among nitrate ammonifiers.

Among the more frequently represented classes in the phylogeny, the co-existence patterns between *nrfA* and denitrification genes displayed a large variation, ranging from 0 in Clostridia and Coriobacteriia to ca. 90% of assemblies carrying at least one denitrification gene in Anaerolineae and Ignavibacteria, and complete denitrifiers were predominantly found among Bacilli (Fig. 2b). Genomes with *nrfA* and genes coding for a nitric oxide (*nor*) or nitrite reductase (mainly *nirK*) were most common and evenly distributed across the phylogeny, whereas those with the N$_2$O reductase gene (particularly *nosZ* clade II) were mainly restricted to Anaerolineae, Bacilli, Ignavibacteria and various lineages of the CXXCH clade (Supplementary Fig. 2). In the *onr* assemblies, members of Gammaproteobacteria and diverse classes of Desulfobacterales (mainly Desulfuromonadia) dominated and the co-existence patterns were largely dominated by the presence of *nor* (Fig. 2d, f).

### Environmental distribution of nitrate ammonifiers

A collection of 1861 globally distributed rhizosphere and soil metagenomes (Fig. 3a) was used to determine the abundance and diversity of *nrfA* and *onr* communities across biomes. Reads corresponding to target gene fragments were identified by mining each metagenome using a hidden Markov model of the reference alignment and candidate sequences were then mapped to the branches of the tree by phylogenetic placement. To account for differences in sequencing depth, *nrfA* and *onr* placement counts were further normalized by the total number of base pairs sequenced in each metagenome (hereafter, 'normalized counts').

The *nrfA* gene was present in all biomes (81 ± 69 counts per Gbp), albeit in different proportions and in some cases with large within-biome variation (Fig. 3b). *nrfA*-ammonifiers were particularly prevalent in rhizosphere and croplands, with intermediate to low phylogenetic diversity (Fig. 3b, c). Tundra soils exhibited the highest phylogenetic diversity, but intermediate normalized *nrfA* counts. Forest ecosystems generally displayed a low abundance and a high diversity, except for tropical and subtropical dry broadleaf forest soils showing high normalized counts and low phylogenetic diversity. By contrast, normalized *nrfA* counts were lower in tropical compared to temperate and subtropical grasslands, savannas and shrublands, but with opposite patterns for phylogenetic diversity. The *nrfA* communities in desert and xeric shrubland soils were characterized by relatively low abundance and diversity. Overall, there was a negative correlation between normalized *nrfA* counts and phylogenetic diversity ($\rho = -0.53$; $p < 0.001$), mainly driven by rhizosphere and cropland communities (Supplementary Fig. 3).

Phylogenetic placement on the reference tree showed that soil *nrfA* communities spanned the entire NrfA region of the phylogeny but the CXXCH clade largely dominated (ca. 90% of the placements)

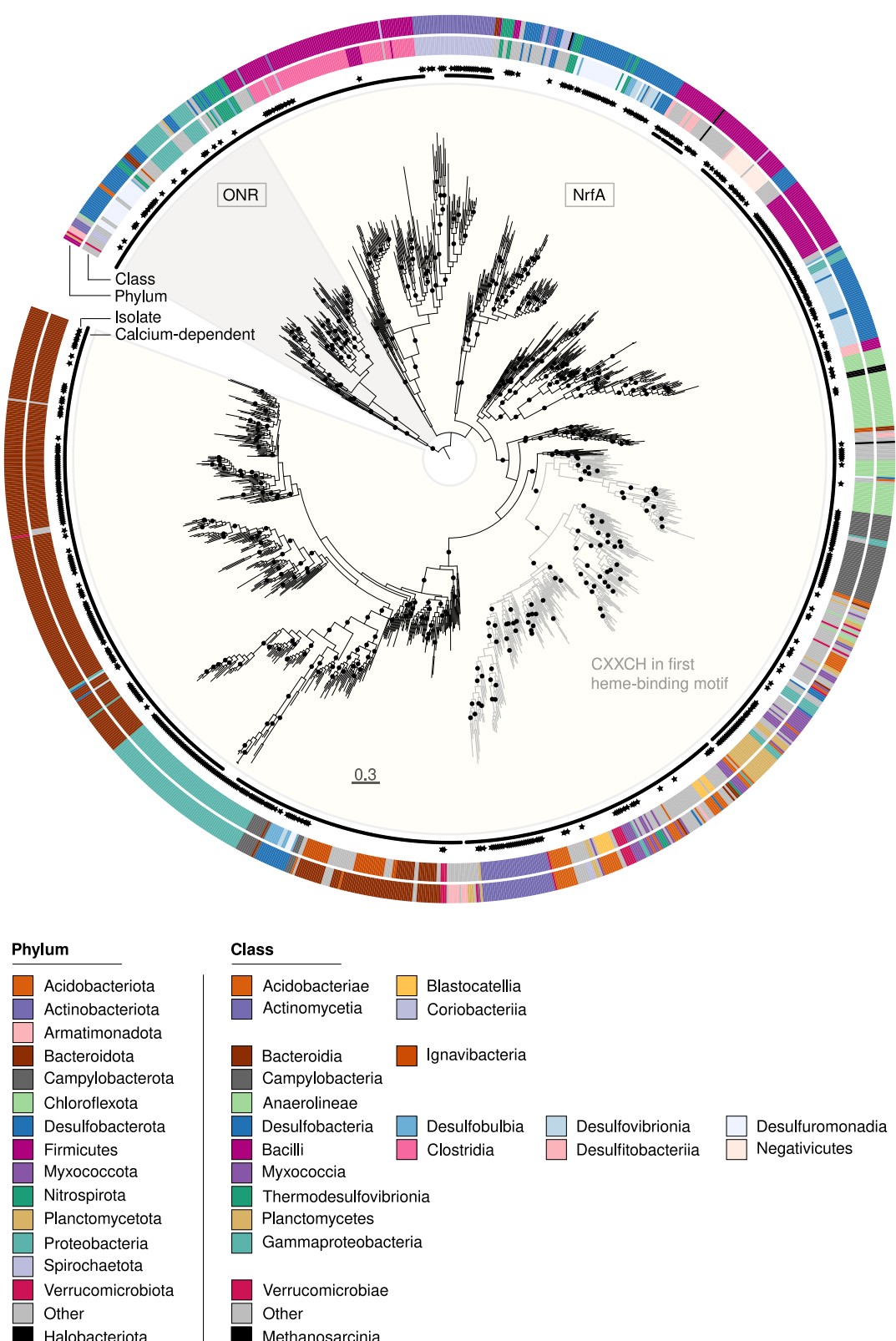

**Fig. 1 | Maximum likelihood phylogeny of 1261 NrfA and ONR sequences from 1218 genome assemblies inferred from the alignment of 350 amino acid positions.** Calcium-dependent sequences are indicated by circles in the inner ring. Sequences obtained from isolates are shown by black stars and the rest are obtained from metagenome-assembled genomes. Taxonomic classification at the phylum and class level of the most abundant classes ($n > 10$, except for the archaeal class Methanosarcinia where $n = 5$) is indicated by the color in the two outer rings and is based on the Genome Taxonomy DataBase. Black circles on the phylogeny show support values (SH-aLRT test ≥ 80% and ultrafast bootstrap ≥ 95%, each threshold corresponding to an estimated confidence level of 95%) and the scale bar denotes the amino acid exchange rate (WAG + R10).

**Phylum**

- Acidobacteriota
- Actinobacteriota
- Armatimonadota
- Bacteroidota
- Campylobacterota
- Chloroflexota
- Desulfobacterota
- Firmicutes
- Myxococcota
- Nitrospirota
- Planctomycetota
- Proteobacteria
- Spirochaetota
- Verrucomicrobiota
- Other
- Halobacteriota

**Class**

- Acidobacteriae
- Actinomycetia
- Bacteroidia
- Campylobacteria
- Anaerolineae
- Desulfobacteria
- Bacilli
- Myxococcia
- Thermodesulfovibrionia
- Planctomycetes
- Gammaproteobacteria
- Verrucomicrobiae
- Other
- Methanosarcinia
- Blastocatellia
- Coriobacteriia
- Ignavibacteria
- Desulfobulbia
- Clostridia
- Desulfovibrionia
- Desulfitobacteriia
- Desulfuromonadia
- Negativicutes

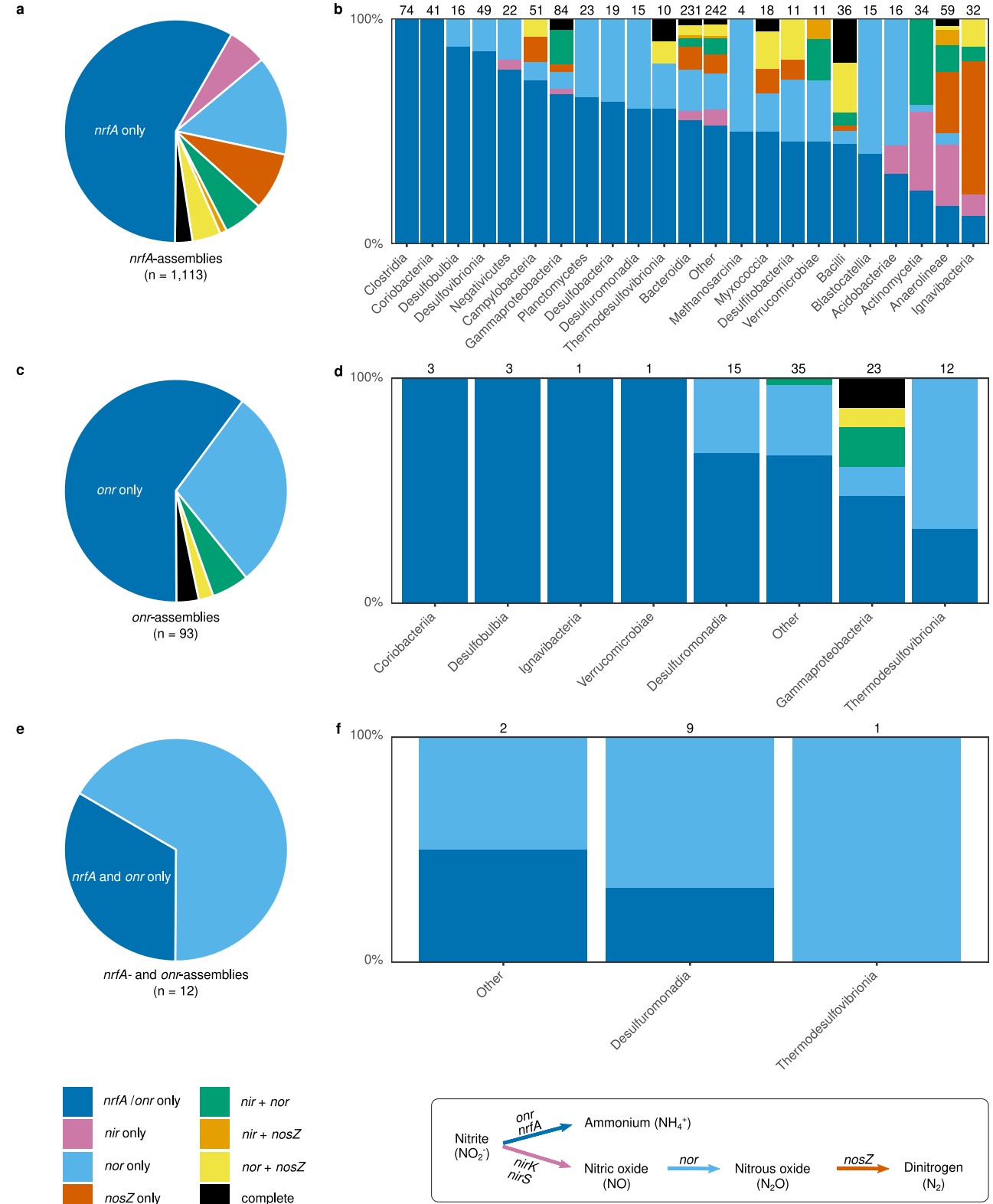

**Fig. 2 | Co-existence of *nrfA*, *onr* and denitrification genes in the 1218 genome assemblies obtained when screening for *nrfA* and *onr*.** The pie charts show the distribution of *nrfA*, *onr* and the denitrification genes *nirK*, *nirS*, *nor* and *nosZ* across (**a**) *nrfA*-only, (**c**) *onr*-only and (**e**) *nrfA−* and *onr−* assemblies. The corresponding bar plots (**b**, **d**, **f**) indicate the distribution of *nrfA*, *onr* and denitrification genes in the classes represented in the phylogeny in Fig. 1. The number of assemblies is indicated above the bar for each class. Classes are ordered according to the proportion of assemblies carrying only *nrfA*/*onr*. Reactions performed by the enzymes encoded by the different genes, with each arrow colored according to the corresponding gene, are indicated at the bottom of the figure.

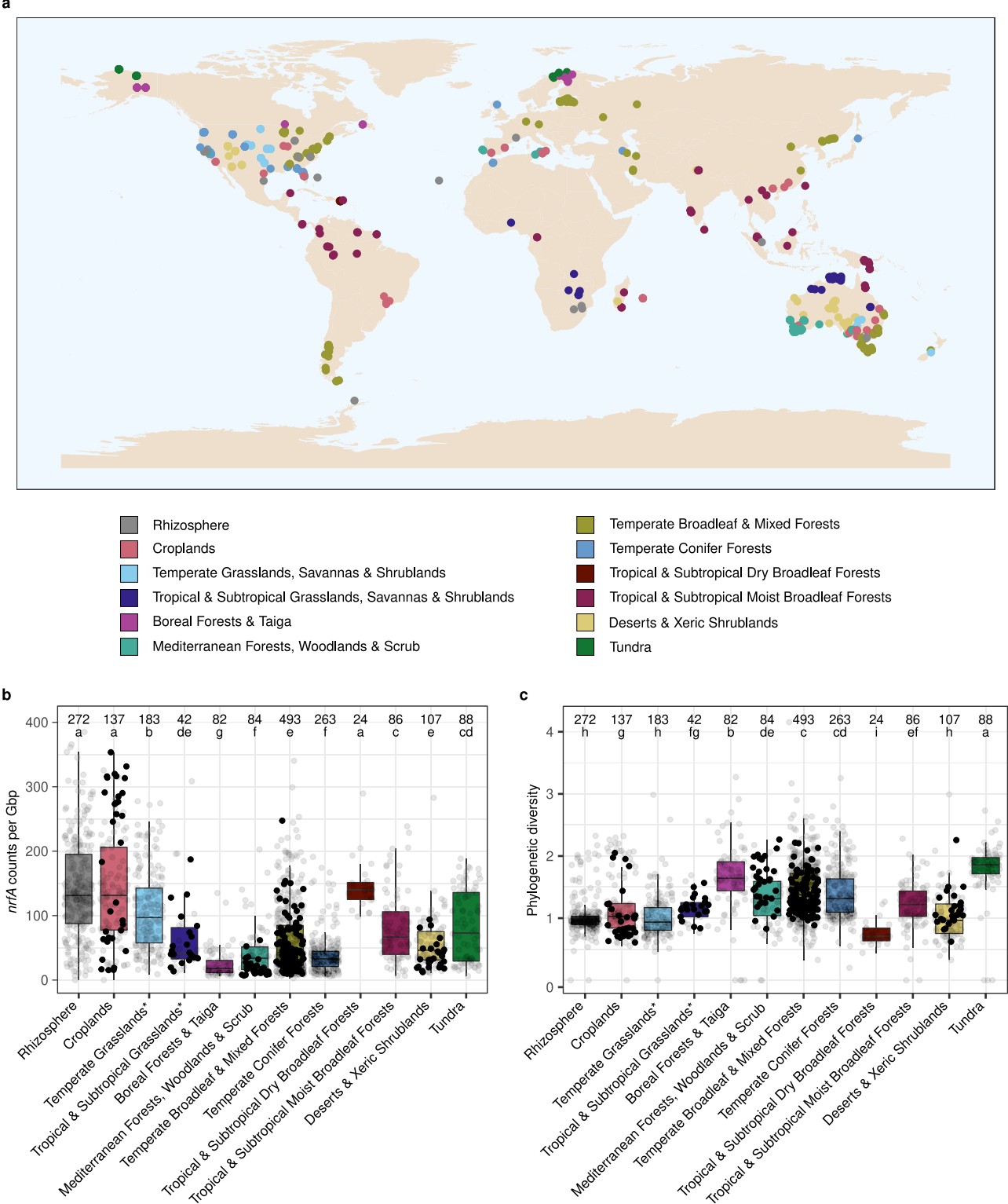

**Fig. 3 | Location of metagenomes and abundance and phylogenetic diversity of *nrfA* across biomes. a** 1861 metagenomes representing 725 sampling sites across the globe. The 35 cropland and 5 rhizosphere metagenomes lacking associated geographic coordinates are not indicated. **b** Normalized *nrfA* counts per biome, calculated as the ratio between *nrfA* counts and the total number of base pairs (Gbp) sequenced in each metagenome (*n* = 1861 metagenomes; Kruskal–Wallis test, H(11) = 641, *P* = 2.68 × 10⁻¹³⁰). **c** Abundance-weighed phylogenetic diversity per biome (*n* = 1861 metagenomes; Kruskal–Wallis test, H(11) = 576, *P* = 2.32 × 10⁻¹¹⁶). Significant differences are denoted with different letters, together with the number of metagenomes representing each biome above the boxplots. Boxes are bounded on the first and third quartiles; horizontal lines represent medians. Whiskers denote 1.5× the interquartile range. Data points corresponding to the metagenomes used in the random forest models are shown as filled circles. *The biome name also includes savannas and shrublands.

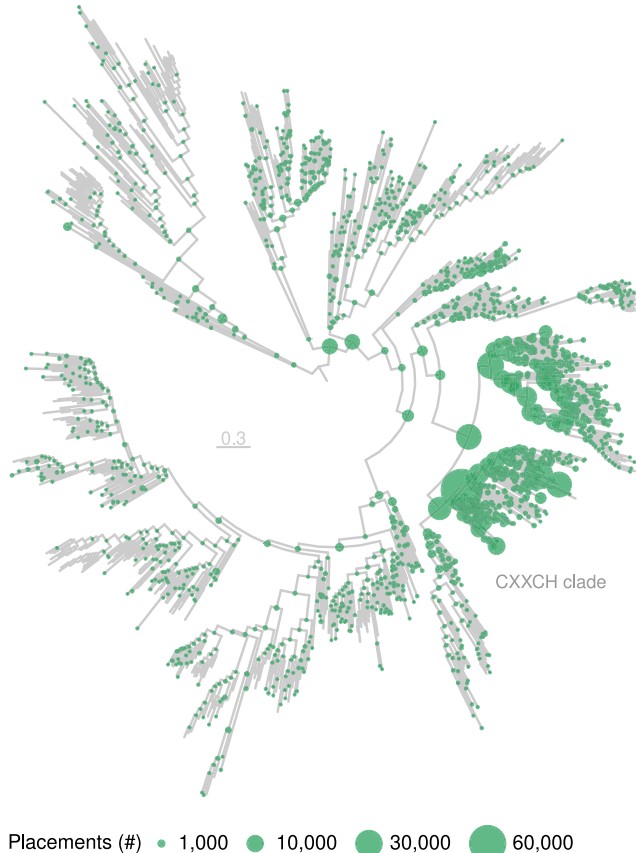

Placements (#)  ● 1,000  ● 10,000  ● 30,000  ● 60,000

**Fig. 4 | Phylogenetic placement of the metagenomic *nrfA* sequence fragments across biomes.** Phylogenetic placement of the metagenomic *nrfA* sequence fragments on the reference tree. The size of the dots is proportional to the number of placements. The scale bar denotes the amino acid exchange rate (WAG + R10). The ONR clade is not shown.

(Fig. 4). Only a few placements were located at or near the tips, indicating that abundant *nrfA*-carrying taxa in soil communities are distantly related to known *nrfA* representatives. Consequently, there was no biome-based discrimination of the *nrfA* communities in the phylogenetically-informed principal component analysis, and most of the variation in community composition was driven by different subsets of the CXXCH clade (Supplementary Fig. 4).

In contrast to *nrfA*, normalized counts of *onr* fragments were low across all biomes, representing ca. 3.5% of the placements (2.7 ± 4.7 counts per Gbp), which prevented the calculation of phylogenetic diversity of *onr*. Similar to *nrfA*, rhizosphere and croplands were among the biomes with the highest *onr* counts, although the greatest abundance was found in tundra (Supplementary Fig. 5). Across the other biomes, *onr* counts were rather similar apart from deserts and xeric shrublands that displayed the lowest abundance. However, due to the low abundance of *onr* there was no statistical difference between *nrfA* only and combined *nrfA* and *onr* counts in any of the biomes (Wilcoxon-Mann–Whitney test, $p > 0.05$; Supplementary Fig. 6). This indicates that nitrate ammonification is dominated by *nrfA*-ammonifiers in terrestrial systems and *onr* counts were therefore not considered in further analyses.

**Relative importance and drivers of NrfA-driven ammonification versus denitrification**

The difference in normalized counts between *nrfA* placements and those of the denitrification marker genes *nirK* and *nirS* was determined to assess the genetic potential for N retention at the community level in each of the 1861 metagenomes (hereafter, '*δnrfA-nir*'). All biomes

exhibited a negative median *δnrfA-nir* with few values > 0, indicating an overall lower genetic potential for NrfA-driven ammonification over denitrification (Fig. 5a). However, median values close to 0 were observed in both tundra and tropical & subtropical dry broadleaf forest soils, which in the latter appeared to be driven by high *nrfA* counts (Fig. 3b). Among forest biomes, the separation by climatic zones observed for the normalized *nrfA* counts (tropical > temperate ≥ Mediterranean > boreal) was not detectable in the *δnrfA-nir* data, suggesting that conditions that favored *nrfA*-ammonifiers were even more favorable to denitrifiers. The median values of *δnrfA-nir* in rhizosphere and cropland communities were at least 2-fold lower than in the other biomes. Among rhizosphere samples, tree (*Citrus* sp. and *Populus* sp.) and perennial grass (*Miscanthus* sp. and *Panicum virgatum*) species displayed higher *δnrfA-nir* than the average, whereas bean (*Phaseolus vulgaris*) and *Brassica* spp. drove *δnrfA-nir* towards higher abundance of denitrifiers. Maize (*Zea mays*) and thale cress (*Arabidopsis thaliana*) displayed *δnrfA-nir* values comparable to the overall mean (Fig. 5b).

Environmental drivers of the normalized *δnrfA-nir* counts were examined using random forest modelling on a subset of the metagenomes, which were selected based on the availability of metadata measured with the same methods across the metagenomes[31] and including relevant factors for NrfA-driven ammonification and denitrification (Table 1). Accumulated local effect plots were used to visualize the differences in prediction of the *δnrfA-nir* along the range of each predictor compared to the mean prediction, with positive values indicating predictions higher than the average, and vice versa. They revealed a non-linear and overall positive relationship between the SOC to nitrate ratio and predicted *δnrfA-nir*, mainly driven by low nitrate content rather than SOC levels (Fig. 6). Soil factors known to affect the activity of the different nitrite reductases also affected *δnrfA-nir*. Calcium, essential for NrfA activity in the abundant CXXCH clade members[29] (Figs. 1 and 4), was associated with an increase in the prediction of *δnrfA-nir* in the ranges 10–25 and 100–125 mM calcium kg⁻¹, whereas copper, crucial for the activity of NirK[32], had the opposite effect (Fig. 6). The *δnrfA-nir* predictions were highest in acidic soils and displayed a *u*-shaped relationship with pH, indicating a threshold at $5.75 < pH < 6.5$ and then increasing *δnrfA-nir* with increasing pH in alkaline (pH > 7.5) soils, which aligns with measurements of nitrate ammonification rates across terrestrial ecosystems[33]. The genetic potential for NrfA-driven ammonification relative to denitrification was predicted to decrease with increasing phosphorus and sulfur (from 5 mg kg⁻¹) concentrations. Biome identity was also identified as an important predictor, even after accounting for other environmental variables. Predictions obtained with the random forest models largely corresponded to the *δnrfA-nir* calculated for the entire dataset, with predictions for croplands and deserts displaying lower relative potential for NrfA-driven ammonification compared to the average prediction, whereas forest and grassland soils had the opposite effect.

## Discussion

Our gene-centric and phylogenetically-informed approach provides a framework for a more accurate understanding of the genetic potential for nitrite reduction in soil communities. We provide evidence that co-existence of *nrfA/onr* and denitrification genes in genomes of ammonifiers is common and phylogenetically widespread, with variation in gene combinations at fine phylogenetic levels and between *nrfA*- and *onr*-assemblies. Those carrying both *nrfA/onr* and *nir/nor* genes can contribute to either N retention or N loss in ecosystems depending on the conditions, whereas those carrying *nosZ* can also act as N₂O sinks[34]. Nevertheless, a higher potential for N₂O production than consumption was observed, mainly due to N₂O production potential linked to NO detoxification. The difference in genetic potential for N₂O production *vs.* consumption was even more pronounced in the CXXCH clade, where most soil-derived *nrfA* reads were placed although not well

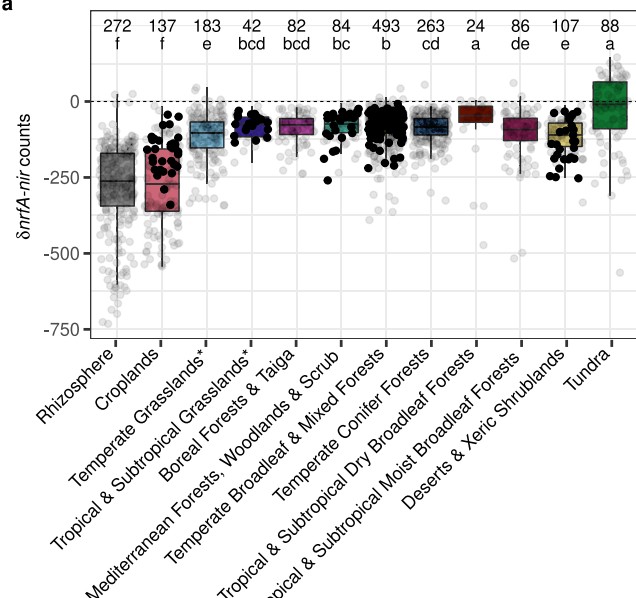

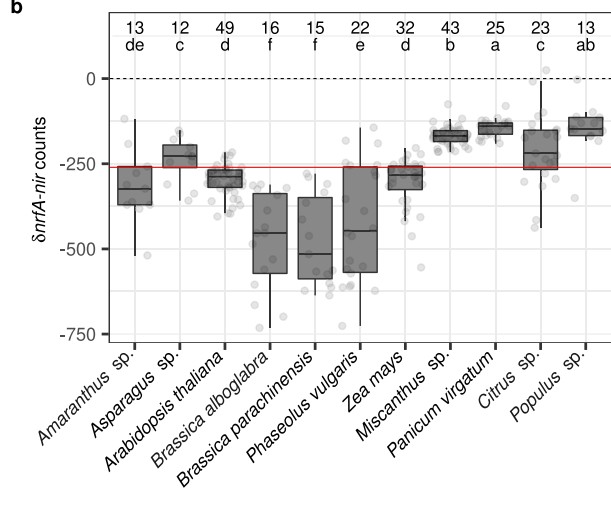

**Fig. 5 | Relative importance of NrfA-driven ammonification and denitrification genetic potential across biomes.** The difference in counts of *nrfA* and *nir* genes normalized by the number of base pairs sequenced (δ*nrfA-nir*) was calculated per metagenome. Positive and negative values indicate a higher potential for NrfA-driven ammonification over denitrification and vice versa. Significant differences are denoted with different letters, together with the number of metagenomes representing each biome. Boxes are bounded on the first and third quartiles; horizontal lines represent medians. Whiskers denote 1.5× the interquartile range. Data points corresponding to the metagenomes used in the random forest models are shown as filled circles. **a** Relative importance of NrfA-driven ammonification and denitrification genetic potential across terrestrial biomes and in the rhizosphere ($n = 1861$ metagenomes; Kruskal–Wallis test, $H(11) = 749$, $P = 1.91 \times 10^{-153}$). **b** Relative importance of NrfA-driven ammonification and denitrification genetic potential in the rhizosphere of host species represented by more than 10 metagenomes ($n = 263$ metagenomes across 11 host species; Kruskal–Wallis test, $H(10) = 174$, $P = 3.69 \times 10^{-32}$). The red line indicates the median δ*nrfA-nir* value across all rhizosphere metagenomes. *The biome name also includes savannas and shrublands.

represented among isolated and genome-sequenced *nrfA*-ammonifiers. Increased efforts aiming at characterizing members of the elusive CXXCH clade is needed to verify these patterns and understand other aspects of their ecology that are relevant to the cycling of N in terrestrial ecosystems. Key questions include whether differences in substrate affinity[35,36] act as a major niche-differentiating factor among ammonifiers, similarly to what is observed in other N transforming groups[37,38], and what role microbial interactions play in providing electron donors (e.g. low molecular-weight organic molecules by fermenters[16]) and nitrite (for ammonifiers lacking the ability to reduce nitrate[36]). Addressing these questions will both facilitate the interpretation of the links between measured rates of nitrate ammonification and ammonifier community composition, diversity, and abundance, and help predicting how environmental factors influence the dynamics of the end-products when nitrate is used as an electron acceptor.

We show that the genetic potential for NrfA- rather than ONR-driven ammonification dominated by a factor of $30 \pm 15$ across all terrestrial biomes, indicating a negligible role of *onr*-ammonifiers for nitrite reduction in soils. Nevertheless, NrfA-driven ammonification potential was in turn much lower than that of denitrification overall in all terrestrial biomes, although the magnitude of the δ*nrfA-nir* counts varied both within and among biomes. Higher nitrate ammonification than denitrification rates have mainly been shown in sulfur-rich or reduced sediments[39–41]. However, we found similar patterns at the genetic level in tundra, as many tundra metagenomes displayed positive δ*nrfA-nir* values. A high abundance of *nrfA*-ammonifiers has indeed been shown to support larger N stocks in these relatively N-limited ecosystems, which indicates that *nrfA*-ammonifiers play a role for N retention[42]. NrfA-driven ammonification could also be a relevant source of substrate for ammonia oxidizers, which potentially drive $N_2O$ emissions rather than denitrifiers in tundra soils[43]. Notably,

managed systems (i.e. croplands and rhizosphere) exhibited significantly lower δ*nrfA-nir* than other soils and this difference was most likely due to fertilizer application[19], as nitrate and phosphorus were among the main drivers of decreased δ*nrfA-nir* in the random forest models. Moreover, lower δ*nrfA-nir* was observed in the rhizosphere of annual plants compared to perennials, which is consistent with the fact that cropping systems with perennial plants are typically characterized by comparatively more favorable conditions for nitrate ammonification, including higher SOC content[44] and higher C/$NO_3^-$ ratios, manifested in lower $N_2O$ production[19]. The effect of SOC on predicted δ*nrfA-nir* rapidly reached a threshold (at ca. 0.8% SOC), indicating a minor role for C quantity, but likely not C quality[35,45]. Instead, the δ*nrfA-nir* was driven by low nitrate content, which aligns with *nrfA*-ammonifiers having a higher affinity than denitrifiers for nitrate or nitrite[46] and with the fact that C availability combined with low nitrate levels creates conditions suitable for nitrate ammonification, as more electrons are transferred per molecule of nitrate reduced[47]. Fertilized soils represent environments where the fate of N is most crucial to control[1] and our findings suggest that integrating N management strategies promoting nitrate ammonification while also increasing soil carbon sequestration represents a promising way to increase N retention, especially in soils with low SOC content. This would reduce nitrate pollution and $N_2O$ emissions, while simultaneously increasing N use efficiency and fertility in agricultural soils.

## Methods
### Generation of reference *nrfA*, *onr* and *nir* phylogenies
Previously published alignments of full-length amino acid NrfA and ONR sequences ($n = 267$ and 27, respectively[27,48]) were used to build hidden Markov models (HMM) using the hmmbuild command implemented in HMMER v. 3.2[49]. The models were used to screen the predicted ORFs in 8131 archaeal and 1,026,048 bacterial assemblies

**Table 1 | Continuous environmental variables associated to the Australian soil metagenomes[31] used in random forest modelling ($n = 227$)**

| Category | Variable | Minimum | Maximum |
|---|---|---|---|
| Soil | Aluminium (mM kg⁻¹) | 0.0 | 31.8 |
| | **Ammonium (mg kg⁻¹)** | 0.0 | 87.0 |
| | Available potassium (mg kg⁻¹) | 15.0 | 855.0 |
| | **Available phosphorus (mg kg⁻¹)** | 2.0 | 193.0 |
| | **Calcium (mM kg⁻¹)** | 0.6 | 158.1 |
| | **Clay (%)** | 0.8 | 65.3 |
| | Conductivity (dS m⁻¹) | 0.0 | 8.9 |
| | **Copper (mg kg⁻¹)** | 0.0 | 32.7 |
| | Iron (mg kg⁻¹) | 1.9 | 1020.2 |
| | Magnesium (mM kg⁻¹) | 0.5 | 60.3 |
| | **Moisture (%)** | 0.0 | 103.3 |
| | **Nitrate (mg kg⁻¹)** | 1.0 | 59.0 |
| | **pH** | 4.0 | 9.6 |
| | Silt (%) | 0.0 | 51.6 |
| | Sodium (mg kg⁻¹) | 2.3 | 6789.6 |
| | **Organic carbon (%)** | 0.1 | 5.9 |
| | **Sulfur (mg kg⁻¹)** | 0.7 | 724.6 |
| | Zinc (mg kg⁻¹) | 0.0 | 32.4 |
| Geomorphology | **Elevation (m)** | 1.0 | 1350.0 |

Variables used in the random forest models (i.e. after the variable selection analysis using pairwise Spearman correlations (Supplementary Fig. 7) and VSURF) are indicated in bold.

The distributions of the gene counts and phylogenetic diversity data in the biomes covered by this dataset are shown by filled circles in Figs. 3 and 5.

available on GenBank in October 2021 for the presence of *nrfA* and *onr* using hmmsearch. An e-value cutoff of 1e-6 was used as a trade-off between reducing the number of false positives before the manual inspection and retaining potentially divergent sequences. The gene sequences were extracted from the assemblies based on genomic coordinates of the HMMER hits to obtain the nucleotide sequence of the candidate NrfA and ONR sequences and were translated to amino acids and dereplicated at 100% identity using CD-HIT v. 4.8.1[50]. The alignments were then manually inspected in ARB v. 7.0[51] for the presence of conserved motifs representing catalytically important residues[11,52]. Due to the existence of homologous multi-heme cytochrome *c* proteins[53], only full-length candidate sequences were retained at this step. Finally, ONR sequences were aligned to the NrfA alignment using HMMER to generate a single alignment, which was then refined by trimming the less conserved and poorly aligned C- and N-terminal regions and by removing columns with >95 % gaps. FastTreeMP v. 2.1.11[54] was used to construct a draft phylogenetic tree and closely related sequences (i.e. very short terminal branch lengths) were manually pruned with the exception of *nrfA* and *onr* copies originating from the same assembly. In the end, the curated data set contained 1261 NrfA and ONR sequences originating from 1218 assemblies (Supplementary Data 1). Selection of the best-fit model of evolution, WAG + R10, and construction of the amino acid-based maximum-likelihood phylogeny were performed with IQ-TREE v. 2.1.3[55,56]. Node support values were calculated using 1000 ultra-fast bootstraps[57] and the Shimodaira-Hasegaw approximate likelihood ratio (SH-aLRT) test[58] with the -bnni option to reduce the risk of overestimating branch supports. The tree was plotted using iTOL v5[59].

Reference phylogenies for NirK and NirS were generated using the same approach. Briefly, previously published alignments of full-length amino acid NirK ($n = 3450$), NirS ($n = 1188$)[12] were used to build HMM

models and search the genome assemblies. For NirK, fungal, plant and protist assemblies (NCBI, accessed in November 2021 and January 2022, respectively) were also included, as well as 18 translated *nirK* sequences from foraminifera transcriptomes[60]. The resulting amino acid alignments containing the candidate sequences were then manually inspected for the presence of conserved motifs in NirK[61] and NirS[62,63]. Two sets of non-target sequences picked up by the hmmsearch were also retained to serve as outgroups in the NirK and NirS phylogenies (various multi-copper oxidases for NirK; NirN, NirF and halophilic archaea NirS-like sequences for NirS). NirK sequences derived from archaeal- or bacterial-like contigs in eukaryotic assemblies were identified using mmseqs2-taxonomy (v. 14[64]) against the UniRef50 database (release 2021_04[65]) and not included in the final tree. After trimming, the alignments were 573 and 471 amino acid long for NirK and NirS, respectively. In the end, 6422 NirK sequences ($n$ outgroup = 367) and 540 NirS sequences ($n$ outgroup = 29) were retained. Selection of the best-fit model of evolution, LG + F + R10 for NirK and LG + F + R9 for NirS, and construction of the amino acid-based maximum likelihood phylogeny were performed with IQ-TREE (Supplementary Figs. 8 and 9).

**Quality check and taxonomic assignment of genome assemblies**
The level of completeness and contamination of each assembly was determined based on the detection of lineage-specific, single-copy genes using BUSCO v. 5.2.0[66]. Assemblies meeting the high quality level standard (completeness >90% and contamination <5%)[67] were retained. The completeness criterium was relaxed for poorly sampled regions of the trees and for eukaryotes. Taxonomic annotations for archaea and bacteria were obtained using GTDB-Tk v. 1.5.0[68] and the reference Genome Taxonomy DataBase r202[69]. For eukaryotes, the taxonomy reported in the NCBI database was used.

**Identification of denitrification genes in genome assemblies**
The presence of denitrification genes (*nor*, *nirK*, *nirS* and *nosZ*) in the *nrfA*- and *onr*- assemblies was determined. A HMM model for NosZ was built by following the approach described above, using a previously published alignment of NosZ sequences ($n = 1689$)[12]. For Nor the alignment from Murali et al.[70] was used. The models were used to search the final set of assemblies and the target sequences were identified by manually inspecting the resulting alignments.

**Construction of a metagenome database and GraftM search**
A database was created by downloading 1861 publicly available soil and rhizosphere metagenomes sequenced using Illumina short-read technology (read length ≥ 150 nt) and consisting of a minimum of 100,000 reads (Supplementary Tables 3 and 4 and Supplementary Data 1). The soil metagenomes were further classified into biomes, with the noncroplands ($n = 1462$) classified following the definition of terrestrial ecoregions proposed by Olson et al.[71]. Biome assignment was performed based on the geographic coordinates of each metagenome, using the 'sp' v. 1.4-6[72], 'rgeos' v.0.5-5[73] and 'rgdal' v. 1.5-23[74] packages in R v. 4.2.0[75]. The geographic location of each sample was plotted using the 'ggspatial' v. 1.1.5[76], 'rnaturalearth' v. 0.1.0[77], 'rnaturalearthdata' v. 0.1.0[78], 'rgeos' v.0.5-5 and 'sf' v. 1.0-7[79] packages.

The presence of *nrfA* and *onr* fragments in the metagenomes was assessed with GraftM v. 0.13.1[80], a gene-centric and phylogenetically-informed classifying tool. Using a custom gene reference package, GraftM identifies target gene fragments in metagenomes using HMMER and places them into a pre-constructed phylogenetic tree using PPLACER[81]. The phylogenetic placement acts as a validation step and ensures that the HMMER hits correspond to the target sequences and not to closely related outgroup sequences. A reference package for GraftM was built using the trimmed alignment and associated phylogeny generated in this study, including both *nrfA* and *onr* sequences. However, the tree was pruned to remove close relatives as

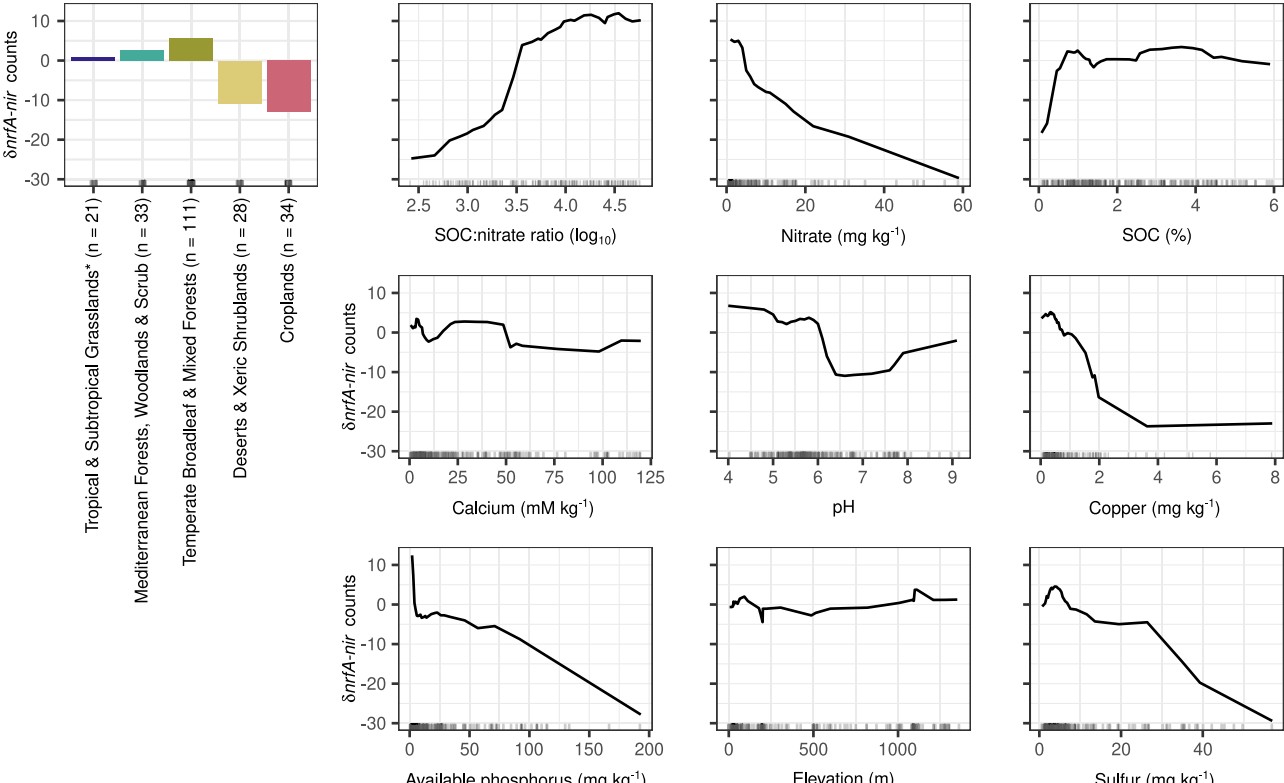

**Fig. 6 | Environmental predictors of the potential competition between *nrfA*-ammonifiers and denitrifiers in soil based on random forest models.** The difference in counts of *nrfA* and *nir* genes normalized by the number of base pairs sequenced (δ*nrfA-nir*) was calculated per metagenome. The analysis was performed on a subset of the metagenomes for which environmental metadata, especially soil properties relevant for nitrate ammonification and denitrification, was available (*n* = 227; Table 1). The number of metagenomes corresponding to each biome is indicated after the biome name. Predictor variables selected by VSURF and biome category were used to generate accumulated local effects plots, which show the differences in prediction of the δ*nrfA-nir* (*y*-axis) compared to the mean prediction along the range of each predictor (*x*-axis), while accounting for potential correlations amongst predictor values. The effect is centred so that the mean effect is zero. The random forest model was built with 500 trees, 2 features considered at each split and a tree depth set to 9 (variance explained: 55%, root mean square error: 40.6). SOC: soil organic carbon. *The biome name also includes savannas and shrublands.

they would increase computation time without increasing the sensitivity of the analyses (final number of tips = 1178). Tree statistics required for running GraftM were calculated using RaxML v. 7.7.2[82] and the tree was re-rooted in iTOL v5[59].

The approach was first validated by fragmenting the aligned region of the *nrfA* and *onr* sequences into 10,000 pieces of 150 nt-long fragments using GRINDER v. 0.5.4[83]. Each set of fragments was then individually processed with GraftM, using default parameters. GraftM provides up to seven placements on the reference phylogeny for each read identified by HMMER and the 'accumulate' command implemented in GAPPA v. 0.8.1[84] was used to find the most likely location in the phylogeny (with --threshold 0.95). This is achieved by accumulating the placement mass (likelihood weight ratio) of the placements of each read upwards (from tips to root), until the accumulated mass reaches the threshold. This means that there is only a 5% likelihood that a placement is not *nrfA* or *onr* (since all placements contributing to the threshold are distributed in clades extending from the branch where the accumulated placement is located). Reads that contained placement mass as both *nrfA* and *onr* were discarded (<2% of the total number of placements). Sensitivity and specificity for *nrfA* were assessed by calculating the fractions of *nrfA* (88%) and *onr* (none) fragments placed into the *nrfA* region of the tree, respectively. Sensitivity and specificity for *onr* were 99 and 100%, respectively. The robustness of this approach to false positives was further examined by fragmenting 86 amino acid sequences corresponding to 8 distant multiheme cytochrome homologs (Cyt c554, HAO, HDH, ihOCC, OTR, MccA, OcwA and OmhA[27]) into 50 amino acid-long fragments (i.e. the

longest peptide which can be predicted from 150 nt reads) and placing them on the reference tree. Only 6% of the fragments were placed in the tree (out of 10,000), with <1% in the CXXCH clade, confirming the limited potential for false positives in this study (Supplementary Fig. 10). Sensitivity and specificity for *nirK* and *nirS* were 88 and 100%, and 87 and 100%, respectively, with gene-specific outgroups for testing the specificity (various multi-copper oxidases for *nirK*; *nirN*, *nirF* and halophilic archaeal *nirS*-like sequences for *nirS*).

GraftM was run on the forward reads of each metagenome with default parameters using the first 150 nt to account for differences in read length between metagenomes. Placements on the reference tree were visualized using the R package 'ggtree' v. 3.4.0[85]. In addition to *nrfA* and *onr*, metagenomes were mined for fragments of nitrite reduction genes involved in denitrification (*nirK* and *nirS*). The resulting placement files were processed with GAPPA as described above. For each metagenome, gene counts were normalized by the number of base pairs (Gbp) sequenced to account for differences in sequencing depth ('normalized counts'). The difference in normalized counts (δ*nrfA-nir*) between *nrfA* and the marker genes for denitrification *nirK* and *nirS* was calculated as (*nrfA*-(*nirK*+*nirS*))/Gbp, which enables differentiation of samples with high absolute differences in genetic potential for NrfA-driven ammonification and nitrite reduction involving *nir* genes from those with small absolute differences. The use of these metrics is relevant to assess the genetic potential for N retention in metagenomes since both *nrfA* (this study) and *nir*[12] genes are most commonly present in single copy in genomes.

## Statistical analyses

Abundance-weighed phylogenetic diversity[86], which provides a normalized measure of the shared phylogenetic history among taxa occurring in a sample, was calculated for each metagenome using the *nrfA* placements and the 'fpd' command implemented in the guppy suite of tools v. 1.1 (with $\theta = 1$). The composition of the *nrfA* community across biomes was examined using the edge principal component analysis[87] implemented in GAPPA ('edgepca' command). This ordination method is based on the phylogenetic placement of reads on a reference phylogeny and allows for the identification of specific lineages that contribute to the variation in composition between samples. It was performed on metagenomes with at least 20 *nrfA* placements as the algorithm otherwise failed to compute ($n = 1475$).

All statistical analyses were performed in R v. 4.2.0. Differences in normalized *nrfA* and *onr* counts, *nrfA* phylogenetic diversity and $\delta nrfA$-*nir* across biomes were assessed using Kruskal-Wallis tests with multiple comparisons computed according to Fisher's least significant difference and the false discovery rate correction available in the 'agricolae' package v. 1.3.5[88]. Differences between *nrfA* and *nrfA*+*onr* normalized counts within biomes were assessed using Wilcoxon-Mann-Whitney tests. Figures were plotted using the 'ggplot2' package v. 3.3.5[89].

Relationships between environmental variables and $\delta nrfA$-*nir* were determined using random forests, an ensemble machine learning algorithm that is well suited to model non-linear relationships between predictors and response variables and can deal with non-normality and high collinearity among predictors[90]. A subset of the metagenomes was selected based on the availability of soil metadata relevant to nitrate ammonification and denitrification, including pH and nitrate, organic carbon, calcium, and copper content ($n = 227$). All metagenomes within this subset belonged to the 'Biomes of Australian Soil Environments' project and covered large environmental gradients at the continental scale (Table 1 and Fig. 3a). Since the corresponding samples were collected and processed following the same protocols[31], this increased the likelihood to detect relevant ecological patterns. Collinearity among environmental factors was assessed by pairwise Spearman correlations (Supplementary Fig. 7) and only the most relevant variables for the processes in focus were retained in each collinear group ($|r| \geq 0.7$; indicated in bold in Table 1). Random forest based variable selection was performed on the pre-selected environmental factors, supplemented with biome category information, using the 'VSURF' package v. 1.1.0[91] to identify the best predictors for $\delta nrfA$-*nir*. The 'randomForest' package v. 4.7–1[92] was then used to model the relationship between the selected predictors and $\delta nrfA$-*nir*. A grid search was first conducted to find the optimal combination of tuning parameters and the combination corresponding to the best model fit (lowest out-of-bag root-mean-square error) was selected. Results were then visualized using accumulated local effects plots (grid.size = 30) implemented in the 'iml' package v. 0.9.0[93,94].

## Reporting summary

Further information on research design is available in the Nature Portfolio Reporting Summary linked to this article.

## Data availability

The metadata, sequence alignments, HMM models and phylogenetic trees (newick format) generated in this study have been deposited in Zenodo (https://doi.org/10.5281/zenodo.8026657). All genome assemblies and metagenomes used in this study were publicly available and their accession codes are provided in Supplementary Data 1. The alignment for Nor was kindly provided by Ranjani Murali. The Genome Taxonomy database can be accessed at: https://gtdb.ecogenomic.org/.

## Code availability

The bash and R scripts generated in this study have been deposited in Zenodo (https://doi.org/10.5281/zenodo.8026657).

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

## Acknowledgements

The work was supported by the Swedish Research Council Formas (grant 2019-00392 to SH) and a senior career grant to SH from the Swedish University of Agricultural Sciences. Part of the computational work and data storage was enabled by resources in projects NAISS 2021/23-527 and NAISS 2021/22-692 provided by the National Academic Infrastructure for Supercomputing in Sweden (NAISS) at UPPMAX, funded by the Swedish Research Council through grant agreement no. 2022-06725. We thank Ranjani Murali for sharing the reference alignment for Nor.

## Author contributions

SH conceived the study and acquired funding. A.S., C.M.J., G.P. and S.H. developed the methodological approach. A.S., C.M.J. and G.P. built the phylogenies. G.P. performed the metagenome search. A.S. conducted the statistical analyses. A.S. and S.H. wrote the original draft and C.M.J. and G.P. provided critical feedback. All authors read and approved the final version of the manuscript.

## Funding

## Competing interests

The authors declare no competing interests.
