## [Peer Review File · Nature Communications]

Phyloecology of nitrate ammonifiers and their importance relative to denitrifiers in global terrestrial biomesREVIEWER COMMENTS

Reviewer #1 (Remarks to the Author):

The paper “Phyloecology of nrfA-ammonifiers and their relative importance with denitrifiers in global terrestrial biomes” addresses the fate of nitrate in soils by comparing the abundance of genes associated with two competing pathways: denitrification and dissimilatory nitrate reduction to ammonia (DNRA). This is important because whereas the former removes N from soil, the latter retains it. The study opens a window to improved soil management where enhancing soil carbon content could combine carbon sequestration, reduce N₂O emissions and reduce fertilizer needs.

I thought this study was really well designed, executed and described overall. The only design choice I do not understand is the exclusion of octaheme nitrite reductases. Both in denitrification and in DNRA two different genes can perform the key biochemical step in parallel. Denitrifiers can use either nirK or nirS, both work. DNRA organisms can use either nrfA or octoheme nitrite reductase, both work. The latter two are distantly related and the authors have used octaheme nitrite reductase sequences as the outgroup in their nrfA tree. But why do they include both nirS and nirK to estimate denitrification potential but decide to only quantify nrfA and not octaheme nitrite reductase to estimate DNRA potential? This gives an incomplete picture of potential DNRA activity and could invalidate all conclusions/inferences of the random forest model when it comes to understanding the competition between the pathways and the consequences for soil management.

It appears the authors currently use octaheme nitrite reductase to reduce false positive nrfA identifications, but I believe a different solution needs to be found for this, as it will not prevent unrelated sequences with weak homology to both nrfA and octaheme nitrite reductase to be incorrectly identified as nrfA.

Introduction

The authors need to include octaheme nitrite reductase (also produces ammonia) in their analysis or explain why it is not relevant: <https://www.sciencedirect.com/science/article/pii/S0014579307007491>
Nitrate ammonification also yields nitrous oxide emissions, as the authors already mention in the discussion - this should be mentioned in the introduction, see also for example <https://ui.adsabs.harvard.edu/abs/2015EGUGA..17.8017B/abstract>

Methods

L268 After dodging arbitrary cutoff values in annotation pipelines, the authors use an arbitrary cutoff value (e-6) to find NrfA sequences. This arbitrary cutoff value needs to be validated using phylogeny and functional/structural analysis.

L269 why did the sequences need to be translated after finding them with a HMM build from an amino acid alignment?

L277 This needs to be better explained because current description does not enable reproduction: “updated HMM including the NrfA sequences identified in the 2019 search (n = 1,889)”

L293 To enable reproduction of the data, the authors must share the alignments and HMM models of all four proteins (nrfA, nirS, nirK, octaheme nitrite reductase) as online data, including NCBI accession numbers of the used sequences. How do the authors prevent false positives for nirS? For example, both nirF and nirN share homology to nirS?

L300 “Lower quality assemblies corresponded to poorly sampled regions of the tree” Which tree?

The authors do not check for false positives, unrelated sequences with some homology, that would be placed as (relatively) long branches either with ingroup or outgroup sequences. Octaheme cytochrome c may not be an effective attractor for these sequences.

Results

L153 “Most placements were located on deep branches in the phylogeny, indicating that abundant nrfA-carrying taxa in soil communities are distantly related to known nrfA representatives.” This could also indicate a prevalence of undetected false positives. See my comment in methods section. Do the deeply branching sequences completely align with the reference nrfA sequences?

L163 I find the “ δ nrfA-nir” concept makes the text REALLY hard to read, it would be so much easier if the authors just phrased differences in relative gene abundances more directly, like “nir genes were more dominant in croplands than in forest soils.” Perhaps even nrfA/nir would be more straightforward to understand if text would otherwise become too verbose.

Reviewer #2 (Remarks to the Author):

This manuscript provides a systematic overview of NrfA-ammonifying bacterial populations in terms of the distribution in genomes of genetic potential relative to co-occurrence with genes affiliated with denitrification (nirS, nirK, nor and nosZ). In addition the authors present global biome analysis of the nrfA gene distribution by screening 1861 meta genomes. They then use a differential metric with nirK and nirS gene distribution relative to nrfA in the same biomes to determine whether there is more potential for denitrification or for nrfA ammonification. Both of these analyses are of value and notably excellent additions to the work previously published by this group on denitrification genes. The more notable results are: 1) nrfA genes commonly co-occur in genomes with denitrification genes. Based on the data in Extended data Table 2 and shown in Figure 2, 42% of nrfA containing genomes possessed one or more denitrification genes. 2) The NrfA CXXCH clade dominates the worlds genomes. And 3) the genetic potential for denitrification, based on nirS + nirK abundance in meta genomes compared to nrfA, is greater than nrfA-ammonification potential. These are interesting observations and analyses. My only issue with the manuscript is in regards to the data analysis regarding the potential for N₂O generation vs. N₂O consumption amongst nrfA containing genomes. I will detail this concern along with a few other issues of note below.

1. On Line 116 the authors state that based on their analysis using data shown in Extended Data Table 2, that non-denitrifying nrfA-ammonifiers were more likely to be N₂O producers than consumers. The problem with this statement is how the authors made their calculations. For calculating the N₂O

production potential, they counted all genomes that had nor only and those that had nir + nor genes. When divided by the total "non-denitrifiers" , the authors get 52%. The issue is that having only a nor gene does NOT make the organism a N2O producer. Only the combination of nir + nor would give an organism the potential to produce N2O. It should be noted that it is not unusual for genomes of many non-denitrifying organisms to have nor genes. They are probably needed for detoxification. They are not contributing to N2O production in any real way. I believe the authors should leave out the genomes with nrfA and nor only has having any meaningful contribution to N2O generation potential. When doing so the percentage of N2O producers falls to 14.6% in all the non-denitrifying nrfA genomes evaluated. The N2O consumers amongst the same group, based on nosZ presence is >31%. It appears the authors didn't include the nosZ + nir subgroup as N2O consumers, and it is not clear why they didn't. It is interesting, that these percentages are much closer in the CXXCH clade, with 28.8% and 20.3% of the non-denitrifier subset associated with N2O consumers and producers, respectively. In my opinion, this illustrates that the statement made on line 116 is not valid. I think the authors just need to modify their analysis in a more physiologically reasonable way. It is clear that among NrfA-ammonifiers, there are potentially those that are N2O consumers and those that are N2O producers. It is not necessary to make a statement about one be significantly greater than the other. By the way, one other observation not noted by the authors, is that the percentage of genomes with any denitrification related genes increases from 42% for all nrfA containing genomes to 59.5% of the nrfA genomes in the CXXCH clade.

2. In Fig. 2b "Other bacteria" are one of the designated classes with 241 representative sequences. How was this designation determined? And is it valid to lump these others into a single group?

3. Figure 4b. This might be due to a limitation in my understanding, but I am not sure I see the benefit of Fig. 4b. Doesn't it show exactly what is expected based on what is shown in Fig 4a?

4. Ln 173-174: I am not sure I follow how more genetic potential presence necessarily translates to "more favorable" conditions being present for denitrifiers vs. NrfA-ammonifiers. Genetic potential does not necessarily correspond to observed functional expression.

5.. Ln 201-204: If the the predictions largely corresponded to the entire dataset, why did the grassland and forest soils show the opposite effect in the random forest model? This would seem to not correspond to the larger dataset.

6. Ln 211-212: Here the authors assert that in the Sanford et al. (2012) paper it is stated : "that genetic capacity for N2O reduction is widespread among nrfA-ammonifiers.". This is a completely inaccurate statement. This reference is about Clade II NosZ and not at all about nrfA. It does make a statement that, among the Clade II nosZ containing genomes evaluated, many of them possessed nrfA genes. This reference makes no claim about evaluating genomes with nrfA, other than those that also had nosZ. What is troubling here is that the authors make an "In contrast... " statement here comparing statements (not really made) in this previous reference to their current results. They go on on line 213 to indicate that nrfA ammonifiers are more likely to be N2O producers than consumers. As noted in my number 1 comment above, I have concerns about how they made this determination.

Reviewer #1 (Remarks to the Author):

The paper “Phyloecology of *nrfA*-ammonifiers and their relative importance with denitrifiers in global terrestrial biomes” addresses the fate of nitrate in soils by comparing the abundance of genes associated with two competing pathways: denitrification and dissimilatory nitrate reduction to ammonia (DNRA). This is important because whereas the former removes N from soil, the latter retains it. The study opens a window to improved soil management where enhancing soil carbon content could combine carbon sequestration, reduce N₂O emissions and reduce fertilizer needs.

I thought this study was really well designed, executed and described overall. The only design choice I do not understand is the exclusion of octaheme nitrite reductases. Both in denitrification and in DNRA two different genes can perform the key biochemical step in parallel. Denitrifiers can use either *nirK* or *nirS*, both work. DNRA organisms can use either *nrfA* or octoheme nitrite reductase, both work. The latter two are distantly related and the authors have used octaheme nitrite reductase sequences as the outgroup in their *nrfA* tree. But why do they include both *nirS* and *nirK* to estimate denitrification potential but decide to only quantify *nrfA* and not octaheme nitrite reductase to estimate DNRA potential? This gives an incomplete picture of potential DNRA activity and could invalidate all conclusions/inferences of the random forest model when it comes to understanding the competition between the pathways and the consequences for soil management.

A: We agree that not including counts of *onr*, coding for the octoheme nitrite reductase, may give an incomplete picture of nitrate ammonification vs denitrification genetic potential patterns across terrestrial environments if they are common and we thank the reviewer for the suggestion to investigate this. This was addressed in three ways. First, we generated an ONR-specific HMM (using the alignment from Soares et al. Mol Biol Evol 2022 as reference) to search the genome assemblies and update the phylogeny with additional ONR sequences. Second, we screened all *onr*-assemblies for denitrification genes similar to what was done for *nrfA*-assemblies. Third, we included the metagenomic *onr* counts in our analyses (see details below).

We searched the 8,131 archaeal and 1,026,048 bacterial assemblies using an ONR-specific HMM. Candidate sequences were processed as described for NrfA and we obtained 106 full-length ONRs, instead of 84 in the original tree. These sequences were used to generate a new phylogenetic reconstruction of NrfA and ONR sequences (see updated Figure 1 and text in the

methods 1. 308-335). The *onr*-carrying assemblies were also screened for the presence of multiple *onr/nrfA* copies and for denitrification genes (see revised Fig. 2 and revised Supplementary Figs. 1 and 2). The results are described in the revised results section (l. 92-100, l. 114-117, l. 131-133, l. 147-150).

In the analysis of the metagenomes, we found that placements of *onr* gene fragments in the ONR part of the phylogeny represented ca. 3.5 % of the placements across all biomes (2.7 ± 4.7 counts per Gbp compared to 81 ± 69 *nrfA* counts per Gbp). The distribution of *onr* fragments in the different biomes is now shown as Supplementary Figure 5 (new figure). This shows that *onr* is 30 ± 15 times less abundant than *nrfA* across all terrestrial biomes. Moreover, there was no statistical difference between *nrfA* only and combined *nrfA* and *onr* counts in any of the individual biomes (see figure below, now included as Supplementary Figure 6 (new figure) in the revised manuscript), indicating that nitrate ammonification potential is largely dominated by *nrfA*-ammonifiers in terrestrial systems.

Supplementary Figure 6. Comparison between *nrfA* and combined *nrfA* and *onr* counts per biome, normalized by the total number of base pairs (Gbp) sequenced in each metagenome. There was no difference between *nrfA* and *nrfA+onr* counts within biomes (Wilcoxon-Mann-Whitney test, $p > 0.05$). Boxes are bounded on the first and third quartiles; horizontal lines represent medians. Whiskers denote 1.5x the interquartile range.

This new information has been added to the text: “In contrast to *nrfA*, normalized counts of *onr* fragments were low across all biomes, representing ca. 3.5 % of the placements (2.7 ± 4.7 counts per Gbp), which prevented the calculation of phylogenetic diversity of *onr*. Similar to *nrfA*, rhizosphere and croplands were among the biomes with the highest *onr* counts, although the greatest abundance was found in tundra (Supplementary Fig. 5). Across the other biomes, *onr* counts were rather similar apart from deserts and xeric shrublands that displayed the lowest abundance. However, due to the low abundance of *onr* there was no statistical difference between *nrfA* only and combined *nrfA* and *onr* counts in any of the biomes (Wilcoxon-Mann-Whitney test, $p > 0.05$; Supplementary Fig. 6).” (l. 187-196).

Because of the low number of *onr* detected in the metagenomes and the lack of statistical difference between *nrfA* only and combined *nrfA* and *onr* counts in any of the individual biomes we decided to not consider the *onr* in the random forest analyses. Indeed, the sum of *onr* and *nrfA* counts would not affect the observed patterns and conclusions about the relative importance of ammonification and denitrification potential in terrestrial biomes. For clarity, we specified in the abstract and throughout the text that the genetic potential for NrfA-driven ammonification, and not nitrate ammonification, was compared to that of denitrification. Nevertheless, we modified the title of the manuscript to indicate the broader take on nitrate ammonification after including the analysis of *onr*: Phyloecology of nitrate ammonifiers and their relative importance with denitrifiers in global terrestrial biomes.

In addition, we have made a correction to the text by removing l. 201-205, as *nir* counts included fungal *nirK*.

It appears the authors currently use octaheme nitrite reductase to reduce false positive *nrfA* identifications, but I believe a different solution needs to be found for this, as it will not prevent unrelated sequences with weak homology to both *nrfA* and octaheme nitrite reductase to be incorrectly identified as *nrfA*.

A: We agree with the reviewer that the approach described in the manuscript does not ensure that sequences with weak homology to NrfA and ONR are not incorrectly identified as NrfA. We thus performed additional analyses. We generated amino acid sequence fragments using GRINDER corresponding to 8 distant multiheme cytochrome homologs: Cyt c554, HAO, HDH, ihOCC, OTR, MccA, OcwA and OmhA ($n = 86$; Soares et al. Mol Biol Evol 2022). We chose to fragment the sequences in 50 amino acid lengths because this is the maximum

sequence length for a 150 nt read, which corresponds to the setting used in the metagenome analysis with graftM.

Supplementary Figure 10. Phylogenetic placement of 86 shredded (50 aa) NrfA and ONR homologs on the phylogeny (Cyt c554, HAO, HDH, ihOCC, OTR, MccA, OcwA and OmhA). Only 6 % of the fragments were placed in the tree (out of 10,000). The scale bar denotes the amino acid exchange rate (WAG+R10). The tree was inferred from the alignment of 350 amino acid positions.

Only 6 % of the fragments (out of 10,000) were placed in the reference tree, showing that our approach was highly specific. Importantly, there were < 1 % placements in the CXXCH clade (which is where 90% of the environmental *nrfA* reads were placed), showing that false positives corresponding to weak NrfA homologs are unlikely to affect the results of this study. We provide the placements on the reference tree as Supplementary Fig. 10 and added the following information to the methods: “The robustness of this approach to false positives was further

examined by fragmenting 86 amino acid sequences corresponding to 8 distant multiheme cytochrome homologs (Cyt c554, HAO, HDH, ihOCC, OTR, MccA, OcwA and OmhA (Soares et al., 2022)) into 50 amino acid-long fragments (i.e. the longest peptide which can be predicted from 150 nt reads) and placing them on the reference tree. Only 6 % of the fragments were placed in the tree (out of 10,000), with < 1 % in the CXXCH clade, confirming the limited potential for false positives in this study (Supplementary Fig. 10).” (l. 428-434).

Introduction

The authors need to include octaheme nitrite reductase (also produces ammonia) in their analysis or explain why it is not relevant.

A: Agree, we have included information about the octaheme nitrite reductase in the introduction (l. 50-53, l. 60-61, l. 71-72, l. 82, and l. 85) and in our analyses, as indicated in the answers to the specific comments above.

Nitrate ammonification also yields nitrous oxide emissions, as the authors already mention in the discussion - this should be mentioned in the introduction.

A: This information has been added to the introduction: “By contrast, only small amounts of N₂O have been detected from isolates performing nitrate ammonification (Stremińska et al., 2012; Mania et al., 2014; Heo et al., 2020) and the process results in the retention of N via the binding of ammonium to negatively charged surfaces in the soil.” (l. 38-41).

Methods

L268 After dodging arbitrary cutoff values in annotation pipelines, the authors use an arbitrary cutoff value (e-6) to find NrfA sequences. This arbitrary cutoff value needs to be validated using phylogeny and functional/structural analysis.

A: Phylogenetic and structural analysis (e.g. examination of conserved motifs at catalytic sites) was performed to ensure that potentially divergent NrfA sequences were obtained while removing false positives through inspection of phylogenetic trees. While we agree that e-6 is arbitrary, we would argue that from a practical perspective this is a very high threshold value especially for searching using an HMM, which by default is more sensitive for picking up distant homologues. For contrast, Soares et al., 2022 used a BLASTp threshold of e-50 for each group of multiheme cytochromes (MHC) investigated in their analysis. To be thorough, we also checked the e-values of non-ONR/NrfA hits by searching the collection of MHC groups in the dataset of Soares et al., 2022 with the HMM used in our initial search. We found that the

OcwA and OmhA groups would be detected at this threshold (highest e-value 8.6×10^{-9}), whereas other more distant homologs had e-values above this threshold. This confirms that distant homologues were present in our initial search, which were filtered out in the subsequent validation steps resulting in ca. 18,000 sequences being retained after filtering and dereplicating at 100% identity (vs ca. 600,000 in the hmmersearch output). In the manual inspection, we considered only full-length sequences displaying all conserved motifs representing catalytically important residues specific to NrfA (ca. 16,000). Phylogenetic reconstruction was used to confirm that candidate NrfA and ONR sequences formed separate clusters and a representative set of 1,155 and 106 NrfA and ONR sequences was selected for downstream analyses, respectively, representing ca. 8% of the post-cutoff set.

This has now been clarified in the text: “An e-value cutoff of $1e^{-6}$ was used as a trade-off between reducing the number of false positives before the manual inspection and retaining potentially divergent sequences.” (l. 314-316). Further, the sentence on arbitrary cut-offs in the introduction has been deleted.

L269 why did the sequences need to be translated after finding them with a HMM build from an amino acid alignment?

A: In our script, we extracted the gene sequences based on genomic coordinates of the HMMER hits to obtain the nucleotide sequence of the candidate NrfA in our database. The nucleotide sequence was then translated to amino acid for manual inspection of conserved motifs specific to NrfA. This has now been clarified in the text: “The gene sequences were extracted from the assemblies based on genomic coordinates of the HMMER hits to obtain the nucleotide sequence of the candidate NrfA and ONR sequences and were then translated to amino acids and dereplicated at 100 % identity using CD-HIT v. 4.8.1 (Fu et al., 2012) (l. 316-319).

L277 This needs to be better explained because current description does not enable reproduction: “updated HMM including the NrfA sequences identified in the 2019 search (n = 1,889)”

A: The description of how *nrfA*, *onr* and *nir* phylogenies were generated has been reformulated for more clarity (l. 308-335). All material needed to reproduce the analyses presented in this study will be available on Zenodo (<https://doi.org/10.5281/zenodo.8026657>). This includes NCBI accession numbers of the genome assemblies, amino acid alignments, HMM models and

phylogenetic trees (newick format), as well as bash and R scripts used for the analyses and the figures.

L293 To enable reproduction of the data, the authors must share the alignments and HMM models of all four proteins (*nrfA*, *nirS*, *nirK*, octaheme nitrite reductase) as online data, including NCBI accession numbers of the used sequences. How do the authors prevent false positives for *nirS*? For example, both *nirF* and *nirN* share homology to *nirS*?

A: Sensitivity and specificity for *nirK* and *nirS* were 88 and 100 %, and 87 and 100%, respectively, with gene-specific outgroups for testing the specificity (various multi-copper oxidases for *nirK*; *nirN*, *nirF* and halophilic archaeal *nirS*-like sequences for *nirS*). This is indicated in the methods section (l. 434-437). As mentioned above, all material needed to reproduce the analyses presented in this study will be available on Zenodo upon acceptance (<https://doi.org/10.5281/zenodo.8026657>).

L300 “Lower quality assemblies corresponded to poorly sampled regions of the tree” Which tree?

A: This sentence refers to the NrfA/ONR phylogeny. This has now been clarified in the text: “Lower quality assemblies corresponded to regions of the NrfA/ONR phylogeny for which high quality assemblies were not available” (l. 372-373).

The authors do not check for false positives, unrelated sequences with some homology, that would be placed as (relatively) long branches either with ingroup or outgroup sequences. Octaheme cytochrome c may not be an effective attractor for these sequences.

A: See our answer above.

Results

L153 “Most placements were located on deep branches in the phylogeny, indicating that abundant *nrfA*-carrying taxa in soil communities are distantly related to known *nrfA* representatives.” This could also indicate a prevalence of undetected false positives. See my comment in methods section. Do the deeply branching sequences completely align with the reference *nrfA* sequences?

A: We acknowledge that the phrasing we used was misleading because the basal branches of the CXXCH clade cannot be considered “deep” in the phylogeny. We thus reformulated the text and also added that ca. 90 % of the placements were found in the CXXCH clade:

“Phylogenetic placement on the reference tree showed that soil *nrfA* communities spanned the entire NrfA region of the phylogeny but the CXXCH clade largely dominated (ca. 90% of the placements) (**Fig. 4**). Only a few placements were located at or near the tips, indicating that abundant *nrfA*-carrying taxa in soil communities are distantly related to known *nrfA* representatives.” (l. 177-182).

Moreover, the counts were based on accumulated placements, as implemented in GAPP, to find the most likely clade of each read. This was achieved by accumulating the placement mass (likelihood weight ratio) of the placements of each read upwards (from tips to root), until the accumulated mass reaches the threshold (0.95). In practice, this means that there is only a 5% likelihood that a placement is not *nrfA* (since all placements contributing to the threshold are distributed in clades extending from the branch where the accumulated placement is located). Reads that contained placement mass in both in and outgroup were discarded. We added this information in the methods (l. 416-421): “This is achieved by accumulating the placement mass (likelihood weight ratio) of the placements of each read upwards (from tips to root), until the accumulated mass reaches the threshold. This means that there is only a 5 % likelihood that a placement is not *nrfA* or *onr* (since all placements contributing to the threshold are distributed in clades extending from the branch where the accumulated placement is located). Reads that contained placement mass in as both *nrfA* and *onr* outgroup were discarded (< 2 % of the total number of placements).”

To address the reviewer’s comment and assess the prevalence of potential false positives due to differences in alignment, we checked whether there were differences in the distribution of basal vs non-basal CXXCH placements along the reference alignment (placements on basal CXXCH branches represented about 20% of the placements in this clade). The distribution showed on the figure below does not display any clear pattern, suggesting that basal CXXCH placements are unlikely to be false positives but simply represent sequences distantly related to known representatives.

Density plot showing the distribution of basal and non-basal CXXCH placements along the reference *NrfA* alignment (starting position). The inset trees show the location of basal and non-basal branches on the tree.

L163 I find the “ $\delta nrfA-nir$ ” concept makes the text REALLY hard to read, it would be so much easier if the authors just phrased differences in relative gene abundances more directly, like “*nir* genes were more dominant in croplands than in forest soils.” Perhaps even *nrfA/nir* would be more straightforward to understand if text would otherwise become too verbose.

A: We decided to use a normalized delta rather than the *nrfA/nir* ratio to analyze the potential for nitrate ammonification vs denitrification because the latter does not reflect absolute differences in the abundance of the genes (i.e. a ratio of 5 may be obtained with 10 *nrfA*/2 *nir* or 1000 *nrfA*/200 *nir*). This is now specified in the text: “The difference in normalized abundance ($\delta nrfA-nir$) between *nrfA* and the marker genes for denitrification *nirK* and *nirS* was calculated as $(nrfA-(nirK+nirS))/Gbp$, which enables differentiation of samples with high absolute differences in genetic potential for NrfA-driven ammonification and denitrification from those with small absolute differences.” (l. 446-450). Thus, using “*nrfA/nir*” could be misleading for the reader and we fear that phrasing in terms of relative abundance of *nir* and *nrfA* genes would become too verbose (we use “ $\delta nrfA-nir$ ” 27 times in the text).

Reviewer #2 (Remarks to the Author):

This manuscript provides a systematic overview of NrfA-ammonifying bacterial populations in terms of the distribution in genomes of genetic potential relative to co-occurrence with genes affiliated with denitrification (*nirS*, *nirK*, *nor* and *nosZ*). In addition the authors present global biome analysis of the *nrfA* gene distribution by screening 1861 meta genomes. They then use a differential metric with *nirK* and *nirS* gene distribution relative to *nrfA* in the same biomes to determine whether there is more potential for denitrification or for *nrfA* ammonification. Both of these analyses are of value and notably excellent additions to the work previously published by this group on denitrification genes. The more notable results are: 1) *nrfA* genes commonly co-occur in genomes with denitrification genes. Based on the data in Extended data Table 2 and shown in Figure 2, 42% of *nrfA* containing genomes possessed one or more denitrification genes. 2) The NrfA CXXCH clade dominates the world's genomes. And 3) the genetic potential for denitrification, based on *nirS* + *nirK* abundance in meta genomes compared to *nrfA*, is greater than *nrfA*-ammonification potential. These are interesting observations and analyses. My only issue with the manuscript is in regards to the data analysis regarding the potential for N₂O generation vs. N₂O consumption amongst *nrfA* containing genomes. I will detail this concern along with a few other issues of note below.

1. On Line 116 the authors state that based on their analysis using data shown in Extended Data Table 2, that non-denitrifying *nrfA*-ammonifiers were more likely to be N₂O producers than consumers. The problem with this statement is how the authors made their calculations. For calculating the N₂O production potential, they counted all genomes that had *nor* only and those that had *nir* + *nor* genes. When divided by the total "non-denitrifiers", the authors get 52%. The issue is that having only a *nor* gene does NOT make the organism a N₂O producer. Only the combination of *nir* + *nor* would give an organism the potential to produce N₂O. It should be noted that it is not unusual for genomes of many non-denitrifying organisms to have *nor* genes. They are probably needed for detoxification. They are not contributing to N₂O production in any real way. I believe the authors should leave out the genomes with *nrfA* and *nor* only as having any meaningful contribution to N₂O generation potential. When doing so the percentage of N₂O producers falls to 14.6% in all the non-denitrifying *nrfA* genomes evaluated. The N₂O consumers amongst the same group, based on *nosZ* presence is >31%. It appears the authors didn't include the *nosZ* + *nir* subgroup as N₂O consumers, and it is not clear why they didn't. It is interesting, that these percentages are much closer in the CXXCH clade, with 28.8% and 20.3% of the non-denitrifier subset associated with N₂O consumers and producers,

respectively. In my opinion, this illustrates that the statement made on line 116 is not valid. I think the authors just need to modify their analysis in a more physiologically reasonable way. It is clear that among NrfA-ammonifiers, there are potentially those that are N₂O consumers and those that are N₂O producers. It is not necessary to make a statement about one be significantly greater than the other. By the way, one other observation not noted by the authors, is that the percentage of genomes with any denitrification related genes increases from 42% for all *nrfA* containing genomes to 59.5% of the *nrfA* genomes in the CXXCH clade.

A: Nitrous oxide and water are the only product of nitric oxide (NO) oxidation by Nor (Shiro 2012, *Biochim. Biophys. Acta - Bioenerg.*), making organisms carrying *nor* potential N₂O producers even if it is for the purpose of detoxification. Thus, N₂O production will occur irrespective of whether microorganisms use Nor to conserve energy via denitrification or detoxify NO. Further, in a complex community N₂O production can also occur irrespective whether NO is produced by the cell carrying *nor* or by a neighboring microorganism. We have therefore kept the the genomes with *nrfA* and *nor* only when calculating N₂O generation potential.

However, we agree that using the term ‘incomplete denitrifiers’ can be confusing. We thus reformulated the text on the proportion on assemblies carrying *nor* vs *nosZ* by focusing on N₂O production vs consumption (this time also including assemblies with the complete gene set for denitrification) and splitting the ammonifier-assemblies (Further, please note that we included ammonifiers with ONR in the revised manuscript (see comments answers to Reciewer #1)) and show the distribution of *nir*, *nor* and *nosZ* genes in the assemblies harboring *nrfA* and *onr* in the new Supplementary Table 2. The text now reads: “About 42 % of the assemblies harboring *nrfA* but not *onr* contained at least one denitrification gene (*nir*, *nor* or *nosZ*), whereas 13 % carried more than one denitrification gene, with complete denitrifiers accounting for just 2.5 % of the *nrfA*-ammonifiers (Fig. 2a; Supplementary Table 2). These proportions were comparatively higher in the CXXCH clade, except for complete denitrifiers (60 %, 18 % and < 1 %, respectively). Among *nrfA* assemblies with at least one denitrification gene, about 50 % were potential N₂O producers, carrying *nor* but not *nosZ*, whereas only 38 % were potential N₂O consumers, carrying *nosZ* alone or in addition to *nir/nor* (51 and 31 % in the CXXCH clade, respectively). Regarding the *onr*-encoding assemblies, either alone or in combination with *nrfA*, 40 % carried at least one denitrification gene, with the potential for N₂O consumption limited to a few gammaproteobacterial genomes (Figs. 2c,e). This suggests a

higher genetic potential for N₂O production than consumption among nitrate ammonifiers.” (l. 121-137).

We did the calculations using the number of assemblies with at least one denitrification gene and not the total number of assemblies because the latter is biased for specific classes, particularly when considering the entire phylogeny (Fig. 2b). Following the reviewer’s suggestion, the percentage of assemblies with at least one denitrification gene in the CXXCH clade is now also indicated in the text (see above; l. 124-131).

2. In Fig. 2b "Other bacteria" are one of the designated classes with 241 representative sequences. How was this designation determined? And is it valid to lump these others into a single group?

A: We detected the presence of *nrfA* and *onr* in 48 phyla but only show the classes represented by at least 10 assemblies in our database in Figs. 1 and 2b,d,f for clarity (except for the archaeal class Methanosarcinia where n = 5). The total database with number of assemblies per phylum is presented in Supplementary Table 1. More details on each of the assemblies (including NCBI accession numbers, taxonomic assignment according to GTDB, denitrification gene content and BUSCO quality score) will be made available on Zenodo upon publication (<https://doi.org/10.5281/zenodo.8026657>).

3. Figure 4b. This might be due to a limitation in my understanding, but I am not sure I see the benefit of Fig. 4b. Doesn't it show exactly what is expected based on what is shown in Fig 4a?

A: We agree with the reviewer and have moved Fig. 4b to the supplementary material (now Supplementary Fig. 4). We decided to keep the figure since the reader could wonder whether there were visible differences in community composition between some of the biomes.

4. Ln 173-174: I am not sure I follow how more genetic potential presence necessarily translates to "more favorable" conditions being present for denitrifiers vs. NrfA-ammonifiers. Genetic potential does not necessarily correspond to observed functional expression.

A: We agree with the reviewer that genetic potential does not necessarily correspond to observed functional expression or activity rates. However, the predictors identified as more favorable for *nrfA* vs *nir* in the random forest analysis (e.g. increasing C:NO₃⁻) align with results from enrichment and pure cultures (Stremińska et al. 2012, Env Microbiol; Kraft et al. 2014, Science; van den Berg et al. 2015, ISME J; Yoon et al. 2015, ISME J), as well as site

specific studies where the effects of different factors on process rates were measured/modelled (Putz et al. 2018, Soil Biol Biochem; Pandey et al. 2019, Soil Biol Biochem; Luo et al. 2020, Soil Biol Biochem). This indicates that gene abundances are a reasonable proxy to identify important drivers for the competition between groups performing these two processes.

5.. Ln 201-204: If the predictions largely corresponded to the entire dataset, why did the grassland and forest soils show the opposite effect in the random forest model? This would seem to not correspond to the larger dataset.

A: Grassland and forest soils did not show the opposite effect, but rather tended to have lower $\delta nrfA-nir$ values compared to the cross-biome mean, and this is what is shown in the first panel of figure 5. More generally, accumulated local effects plots (Fig. 6) show the differences in prediction of the $\delta nrfA-nir$ (y-axis) in the different biomes (x-axis) compared to the mean prediction across biomes (<https://christophm.github.io/interpretable-ml-book/ale.html>). Positive values thus indicate predictions higher than the average, and vice versa. In Fig. 5, deserts and croplands displayed lower $\delta nrfA-nir$ than forests and grasslands, which is consistent with the fact that the RF model predicted lower and higher value than the average for these biomes, respectively. Likewise, the fact that croplands values were predicted to be lower than those of deserts is consistent with patterns in Fig. 5, as were the predictions for grassland (lower) and forests (higher). The text has been revised to provide a clearer explanation of the accumulated local effects plots: "Accumulated local effect plots were used to visualize the differences in prediction of the $\delta nrfA-nir$ along the range of each predictor compared to the mean prediction, with positive values indicating predictions higher than the average, and vice versa." (l. 223-227).

6. Ln 211-212: Here the authors assert that in the Sanford et al. (2012) paper it is stated : "that genetic capacity for N₂O reduction is widespread among nrfA-ammonifiers.". This is a completely inaccurate statement. This reference is about Clade II NosZ and not at all about nrfA. It does make a statement that, among the Clade II nosZ containing genomes evaluated, many of them possessed nrfA genes. This reference makes no claim about evaluating genomes with nrfA, other than those that also had nosZ. What is troubling here is that the authors make an "In contrast..." statement here comparing statements (not really made) in this previous reference to their current results. They go on on line 213 to indicate that nrfA ammonifiers are more likely to be N₂O producers than consumers. As noted in my number 1 comment above, I have concerns about how they made this determination.

A: We deleted the comparison with the study by Sanford *et al.* and the text now reads: “We provide evidence that co-existence of *nrfA/onr* and denitrification genes in genomes of ammonifiers is common and phylogenetically widespread, with variation in gene combinations at fine phylogenetic levels and between *nrfA*- and *onr*-assemblies. Those carrying both *nrfA/onr* and *nir/nor* genes can contribute to either N retention or N loss in ecosystems depending on the conditions, whereas those carrying *nosZ* can also act as N₂O sinks.” (l. 249-254). As indicated above, we think that considering nitrate ammonifiers carrying *nor* as potential N₂O producers is relevant, even when *nor* is used for detoxification only.

REVIEWERS' COMMENTS

Reviewer #1 (Remarks to the Author):

I thank the authors for the way they thoroughly and convincingly addressed my concerns.

Reviewer #2 (Remarks to the Author):

I considered this manuscript to be very good in my initial review. It is now in excellent shape. I am satisfied with the modifications to the manuscript and responses to my concerns.